# Ultrasound Histotripsy on a Viable Perfused Whole Porcine Liver: Histological Aspects, Including 3D Reconstruction of the Histotripsy Site

**DOI:** 10.3390/bioengineering10030278

**Published:** 2023-02-21

**Authors:** Saied Froghi, Andrew Hall, Arif Hanafi Bin Jalal, Matheus Oliveira de Andrade, Layla Mohammad Hadi, Hassan Rashidi, Pierre Gélat, Nader Saffari, Brian Davidson, Alberto Quaglia

**Affiliations:** 1Department of HPB & Liver Transplantation Surgery, Royal Free London NHS Foundation Trust, Pond Street, Hampstead, London NW3 2QG, UK; 2Centre for Surgical Innovation, Organ Regeneration and Transplantation, UCL Division of Surgery & Interventional Sciences, Royal Free Hospital Campus, Pond Street, Hampstead, London NW3 2QG, UK; 3Department of Cellular Pathology, UCL Cancer Institute Royal Free London NHS Foundation Trust, Pond Street, Hampstead, London NW3 2QG, UK; 4UCL Medical School, University College London, 74 Huntley Street, London WC1E 6BT, UK; 5Ultrasonics Group, Department of Mechanical Engineering, Roberts Engineering Building, University College London, Torrington Place, London WC1E 7JE, UK; 6Developmental Biology and Cancer Program, UCL Great Ormond Street, Institute of Child Health, University College London, 30 Guilford Street, London WC1N 1EH, UK; 7Department of Surgical Biotechnology, Division of Surgery and Interventional Science, UCL, Royal Free Hospital Campus, Pond Street, Hampstead, London NW3 2QG, UK

**Keywords:** boiling histotripsy, liver, whole organ perfusion system, pig liver, birefringence, necrosis

## Abstract

Non-invasive therapeutic-focused ultrasound (US) can be used for the mechanical dissociation of tissue and is described as histotripsy. We have performed US histotripsy in viable perfused ex vivo porcine livers as a step in the development of a novel approach to hepatocyte cell transplantation. The histotripsy nidus was created with a 2 MHz single-element focused US transducer, producing 50 pulses of 10 ms duration, with peak positive and negative pressure values of P+ = 77.7 MPa and P− = –13.7 MPaat focus, respectively, and a duty cycle of 1%. Here, we present the histological analysis, including 3D reconstruction of histotripsy sites. Five whole porcine livers were retrieved fresh from the abattoir using human transplant retrieval and cold static preservation techniques and were then perfused using an organ preservation circuit. Whilst under perfusion, histotripsy was performed to randomly selected sites on the live. Fifteen lesional sites were formalin-fixed and paraffin-embedded. Sections were stained with Haematoxylin and Eosin and picro-Sirius red, and they were also stained for reticulin. Additionally, two lesion sites were used for 3D reconstruction. The core of the typical lesion consisted of eosinophilic material associated with reticulin loss, collagen damage including loss of birefringence to fibrous septa, and perilesional portal tracts, including large portal vein branches, but intact peri-lesional hepatic plates. The 3D reconstruction of two histotripsy sites was successful and confirmed the feasibility of this approach to investigate the effects of histotripsy on tissue in detail.

## 1. Introduction

Ultrasound histotripsy is a non-invasive procedure used to mechanically fractionate and liquefy soft tissue without causing thermal damage. Different ultrasound pulsing protocols can be used to cause mechanical tissue damage. The one employed in this study is referred to as boiling histotripsy (BH), where a series of millisecond-long ultrasound pulses are delivered to the treatment site. The nonlinear propagation of ultrasound waves results in the formation of shockwaves at the beam focus. High-amplitude shocks cause rapid and transient tissue heating, resulting in the nucleation of a boiling bubble [1,2,3,4]. The interaction of incident shocks with the boiling bubble causes tissue disruption and liquefaction without significant thermal injury to adjacent tissue. The efficacy of mechanical fractionation, the ablation rate, and the reliability of boiling histotripsy depend on the correct choice of ultrasound frequency, focal peak pressure levels, pulse duration, the pulse duty cycle, and the number of BH pulses. The interplay between these parameters has been studied by many groups and extensively reported [5,6]. Possible applications of BH include the ablation of solid tumours [7,8,9,10], local drug delivery [11,12], and cavity formation for direct intrahepatic cell delivery for cell therapy [13]. We have carried out a series of studies using different US histotripsy protocols to create a nidus in the liver for transplantation of stem cell-derived 3D hepatocytes for the treatment of congenital and acquired liver disease.

The literature on the histological effects of histotripsy on liver tissue is limited and the results may vary with the experimental model used [14]. There have been prior experiments involving bovine liver [3,15,16,17,18]. Simon et al. [16], for example, investigated how changes in static pressure and tissue wetness were shown to affect BH injury. Increasing static pressure resulted in the formation of a mound on the cut surface of liver slices in contrast to the cavity generated at atmospheric pressure. The authors also applied their protocol to an in vivo porcine model and showed that wetting the liver surface targeted by BH with saline or surfactant increased tissue atomization and caused the rupture of the liver capsule in some instances. In experiments by Vlaisavljevich et al. [17], thermal pre-treatment affected liver stiffness and its susceptibility to histotripsy. Variations in the intensity of the trichrome stain depending on pre-treatment temperature were used as indicators of changes in collagen density. Macoskey et al. [18] examined the effect of different histotripsy dosages on collagen I and collagen III fibers on agarose-embedded 4 cm cube bovine liver samples using histochemical stains (Gordon and Sweet and trichrome stains) and showed that the cellular component was destroyed earlier than the extracellular matrix.

Other experiments have been carried out on pig liver. Cavitation cloud histotripsy on ex vivo and in vivo livers [17,19,20] caused complete fractionation of hepatic parenchyma and sharp lesional boundaries but no injury to large hepatic vessels or bile ducts. In a detailed histological examination of livers treated with BH, Wang et al. [21] described variations in tissue damage according to duty cycle. The authors assessed collagen fibers, vessels, bile ducts, and cellular function by NADH-diaphorase staining, protein content, and cellular components by electron microscopy on samples removed from the lesional area. An increase in duty cycles resulted in the formation of a macroscopically visible blanched border of up to 3 mm in width, which corresponded, microscopically, to a perilesional area of partial thermal denaturation with loss of NADH-diaphorase stain, an increased amount of intralesional nuclear debris, and the formation of heat-fixed eosinophilic tissue as the result of thermal coagulation. The damage to the connective tissue framework was visible macroscopically and corresponded to microscopic changes in the configuration of collagen fibers, from fibrillary at lower duty cycles to globular at higher duty cycles, along with a concomitant reduction in the number of small-caliber vessels and bile ducts.

In this paper we provide a detailed description of the histological changes we have observed at 15 BH sites and the surrounding tissue. We have also carried out, for the first time, 3D reconstructions of two additional BH sites.

## 2. Materials and Methods

Five pig livers were obtained fresh from the abattoir. Livers were obtained within 10 min of the termination, which was the period of liver warm ischemia. Using organ transplant retrieval techniques, livers were immediately dissected from the complete set of abdominal viscera, with isolation of the hepatic artery, portal vein, and bile duct (en bloc retrieval of the pig’s abdominal viscera, followed by on-table retrieval of liver and its subsequent on-table flush). The isolation of the liver from the rest of the viscera took a total of 2 min. At the abattoir, back table perfusion and flush of the livers were achieved following retrieval with 1 L of heparin saline solution via the portal vein. Flushing was conducted as fast as possible to drain the liver of any remaining blood (the flush would typically be complete in 10–15 min following cannulation of the portal vein). The livers were packed for transport from the abattoir using static cold storage in preservation with normal saline and stored in crushed ice containers for transport at an average temperature of 5 °C. Upon the arrival of the organ to the laboratory, the livers were placed in an organ bath lying over an ultrasound absorbing layer. The organ was re-perfused in two stages using an organ perfusion circuit with a pump. First, the liver was perfused via the main portal vein with a liter of Soltran organ preservation solution (Baxter Healthcare, Newbry UK), to allow the core temperature of the liver to slowly reach room temperature (ranging from 24 to 30 °C) before the histotripsy was performed. During the histotripsy period, the organ was further perfused with Soltran organ preservation solution to maintain viability. Perfusion occurred via the portal vein, with the perfusate draining via the vena cava into an organ bath. The perfusate was not recycled. During the perfusion, the livers would be placed in an organ bath lying over an ultrasound absorbing layer. The perfusion solution was delivered using a Baxter perfusion pump™ (Baxter Healthcare, Newbry, UK) to achieve a constant flow rate of 350 mL/h, ensuring vessel patency before the histotripsy lesions were created.

The setup of the histotripsy system is outlined below. The US transducer was applied to random positions (total sonication time per lesion was 50 s) on the perfused liver, and focal lesions could be identified by puckering of the capsule of the liver at the histotripsy site. Locations for histotripsy were chosen to include both central and peripheral portions of each liver. The depth of the lesion was estimated to be 0.5 cm below the liver capsule. Given that we used the same transducer, the depth was assumed to remain the same for every lesion. A total of 17 BH lesions were created and submitted for histological examination.

### 2.1. Histotripsy Setup

A 2 MHz single element spherically focused transducer (Sonic Concepts H-148, Bothell, WA, USA) with an aperture size of 64 mm, a radius of curvature of 63.2 mm, and a 22.6 mm central opening was used with a transparent coupling cone (Sonic Concepts C-101, Bothell, WA, USA) filled with degassed, de-ionized water. The transducer was driven by two function generators (Agilent 33220A, Sacramento, CA, USA) in series via a linear radiofrequency (RF) power amplifier (ENI 1040 L, Rochester, NY, USA). The first function generator was set to generate 50 pulses of a 1 Hz square wave, with 1% duty cycle. This triggered the second function generator, which yielded a 2 MHz sinusoidal wave into the RF power amplifier. Therefore, the drive signal into the amplifier was 50 pulses of 10 ms duration, containing 20 k cycles. A power meter (Sonic Concepts 22A, Bothell, WA, USA) was connected between the RF amplifier and the ultrasound transducer, and the electrical power supplied to the transducer was monitored to be approximately 150 W. The pulse-averaged electrical power was 1.5 W (calculated using P_averaged_ = P_peak_ × duty cycle). Assuming a nominal electrical to the acoustic power conversion efficiency of 85% (Sonic Concepts, Bothell, WA, USA), the acoustic peak positive (P+) and negative (P−) pressures at the HIFU focus on liver tissue were P+ = 77.7 MPa and P− = –13.7 MPa, obtained by numerically solving the Khokhlov–Zabolotskaya–Kuznetsov (KZK) parabolic nonlinear wave propagation equation for our input parameters using the HITFU Simulator v2 (Soneson 2017). The simulated acoustic waveforms and peak pressures at the HIFU focus are shown in Figure 1. During the experiments, a polyurethane rubber acoustic absorber (AptFlex F28, Precision Acoustics Ltd., Doschester, UK) was placed under the liver samples to minimize ultrasonic reflections. This protocol was chosen because, according to previous extensive work by the authors [13,22,23] and other groups [3,21,24], it leads to the production of a fully fractionated lesion in liver tissue with minimal thermal denaturation. It was one of our aims to examine this assertion with more advanced histological analysis.

### 2.2. Tissue Sampling

After the application of BH, the lesional site was resected with an approximate margin of 1 cm and bisected using a surgical blade to confirm the lesional site. Resection refers to an excision biopsy with a surgical blade, with an approximate circumferential margin of 1 cm from the core lesion or its surface landmark to the liver capsule dimple. Bisection refers to the slicing of the resected sample in half, cutting through the middle and therefore dividing the lesion into two halves to gain access to the core content of the lesion. Then, a 20 µm single-channel gauge pipette was applied gently to the lesional surface to aspirate the cavity content (results are presented in separate study). Although the protocol was specifically designed for cell harvesting from the core of the cavity in the perfused pig liver model, care was taken to minimize traumatic damage to the tissue. Lesions would only be aspirated if the cavity content were present at the time of bisecting the lesion. For a small number, due to proximity to small vessels, as soon as the cavity was sliced open, the content was washed away due to puncture of the vessels at the site and perfusate leaking through the vessel. These lesions were not aspirated and only biopsied. Both sides of the lesion were then removed in the form of wedge-shaped biopsy samples, which were placed immediately into a single formalin-filled container and submitted for paraffin embedding according to standard procedures.

### 2.3. H&E and Histochemical Stains

A single 4 µm paraffin section from each sample (i.e., containing both wedge-shaped biopsy specimens) was stained with Haematoxylin (Harris Haematoxylin, Shandon, UK) and Eosin (Eosin Y, Shandon, UK) (H&E) using an autostainer (Leica ST5020, Milton Keynes, UK).

One additional section from each sample was stained with picro-Sirius red (SR) stain (Sigma-Aldrich Direct Red 80, Dorset, UK) with the following protocol: (1) deparaffinize sections in 2 changes of xylene; (2) rehydrate through 2 changes of alcohol (70% alcohol and distilled water); (3) apply SR solution to cover entire sections and incubate in humidified chamber for 60 min; (4) after 60 min, rinse slides quickly in 2 changes of Acetic Acid Solution; and (5) rinse slides in absolute alcohol and then dehydrate through 2 changes of alcohol, clear in 2 changes of xylene, and mount with pertex mounting medium.

A further section from each sample was stained for reticulin. The staining procedure was carried out using the Reticulum II stainer kit (Roche catalogue number 860-024) and a Ventana BenchMark Special Stain system.

### 2.4. Three-Dimensional Reconstruction of the Histotripsy Site

Two additional histotripsy specimens were removed intact and entirely sectioned at 4 µm into 368 and 408 serial sections, which were stained with H&E and SR, respectively. In the first sample, the lesion was identified in sections 199 to 228 (cumulative section thickness of 116 µm). In the second specimen, minute lesional sites were identified in the full range of serial sections (1-408). The lesional serial sections were used for 3D reconstruction of the histotripsy sites using the specifically designed method illustrated in Figure 2 with the following steps.

1. Sectioning: Serial sections of formalin-fixed paraffin-embedded (FFPE) liver were cut at 4 µm, and any sections lost or missed were noted at the time of sectioning.

2. Images: The tissue was then stained with H&E in the first model and SR for the second, slides were imaged using a Zeiss AxioCamICc5 and Axiovision v2.8.1 (Zeiss, Oberkochen, Germany), and images were saved in lossless TIFF format (Figure 2A).

3. Mark-up: Each image was marked up using software that allows altering of drawn pixels to a single RBG color code (e.g., Microsoft Paint, Microsoft Corporation, Redmond, WA, USA). Many image-editing software packages can be used for this purpose; however, it is important to use a tool that allows editing the image so that the change to any pixel is to that of a set RGB colour value and all changed pixels are of that value. Many tools will feather or blend the edges of the marked area, which will cause problems later. This was used to define areas of interest, such as portal tracts, capsules, or lesions (Figure 2B), and the structures modelled in the final 3D reconstruction. Each RGB colour code defined an aspect of the final 3D model (e.g., red for blood vessels, green for capsules, blue for lesions).

A saturated blue (RGB, 0,0,255) rather than one similar to that of haematoxylin was used. Critical to this step was the use of lossless image formats as lossy formats, such as .jpeg, can cause image artefacts and loss of colour fidelity, reducing the accuracy of the model. In essence, each image slice has been hand-segmented according to the prevalent tissue morphology.

4. Registration: The images were registered using a custom MATLAB (The Mathworks, Natick, MA, USA) script. This script allows two images of serial sections to overlie each other and the alpha (transparency of the images) to be adjusted to visualize both images at once. Distinct features, such as the liver capsule and the edge of the tissue, can be used to align each image over one another and, when a best fit is found, the images are then automatically cropped and saved, (Figure 2C) forming a stack of registered images.

5. Extracting binary matrices: From the stack of registered images, we created a stack of binary matrices for each different type of feature (i.e., each colour is used in the Mark-up step). A 2D binary matrix is an image composed of solely 1 and 0 s and can be visualized as a black and white image with no greyscale in between. To create the binary matrix from the markup images, we used a second custom MATLAB (The Mathworks, Natick, MA, USA) script in which a colour range is chosen, and the pixels within the range are given a 1, while those outside of the range are given a 0, thus picking out a particular marked up feature. For example, the range chosen for the saturated blue mentioned in ‘Mark-up’ would be inclusive of only the pixels with a RGB value of 0,0,255. The range can be widened to pick out tissue features such as all the collagen in a SR-stained section if the stain is of sufficient quality. Each stack of 2D binary matrices, when combined as a stack into a 3D binary matrix, forms a 3D representation of each marked up feature.

6. Three-dimensional matrix: The segmented data of each morphological feature were subsequently rendered in 3D using an in-house developed MATLAB-script (The Mathworks, San Mateo, MA, USA). From this 3D binary matrix of voxels, a triangle mesh was formed on the outer surface of the voxels and the mesh was saved as a Standard Triangle Language (.stl) file (Figure 2F—collagen). A separate .stl file was created for each feature (e.g., blood vessels, capsules, lesions).

7. Mesh visualization: .stl files are a commonly used file format for 3D meshes. The images in Figure 2F are from Meshlab (Cignoni et al., 2008) open-source software. Meshlab is an open-source software, created and maintained by the Visual Computing Lab, part of the ISTR-CNR in Pisa, Italy (www.meshlab.net, accessed on 30 October 2022). Multiple meshes, one for each feature, were loaded into the 3D space while retaining their initial relative positions to one another and were then textured to create a final 3D model of the tissue (Figure 2G).

## 3. Results

The histotripsy lesion consisted of an area of loss of hepatic plates within the hepatic parenchyma (Figure 3). The core of the lesion consisted of loose eosinophilic material (Figure 4a). Hepatocytes or other cell types could no longer be identified. Minute specks of basophilic material were also present in places in keeping with nuclear debris and scattered nuclei, which appeared to be morphologically intact (Figure 4a inset). These changes were designated as ‘type A lesions’ and were present in all 15 samples. Smaller areas characterized by dense eosinophilia, including eosinophilic clumps, were also present in all 15 samples, usually at the periphery of type A lesions (Figure 4a). These areas were designated as ‘type B lesions’. The perisinusoidal extracellular matrix (ECM) marked by the reticulin stain in type A and type B lesions showed loss and fragmentation of reticulin fibers (Figure 4b). The perilesional hepatocytes and their supporting ECM appeared to be morphologically intact.

In all cases, ambient interlobular connective tissue septa within the histotripsy lesion showed distortion. In 10 out of 15 cases (67%), broken interlobular septa were also identified (Figure 4c–f). The collagen fibers in distorted septa retained their birefringence properties under polarized light (Figure 4d–f). This type of collagen fiber damage was designated as a ‘type C lesion’. Type C lesions often co-localized with type B lesions. In some of the distorted septa, collagen fibers appeared to have condensed into coarser bright red bundles or clumps, showing reduced or no birefringence under polarized light (Figure 4e,f). This type of collagen damage was designated as a ‘type D lesion’. Type D lesions also tended to co-localize with type B lesions.

In all cases, damaged portal tracts were present at the periphery of the BH lesions (Figure 5a), with encasement and/or damage to local small portal biliary and/or vascular structure. In one case, the BH lesion broke the wall of the branch of a portal vein of 800 µm cross sectional diameter (Figure 5b). In 11 (73%) cases, the core lesion included empty spaces. The liver surface capsule appeared intact. The BH lesion was more than 2 mm deep in most (11, 73%) specimens and affected the subcapsular parenchyma in four (27%) specimens. In these four specimens, one or more deeper BH lesions were separated from the subcapsular lesion by intact liver parenchyma.

### Three-Dimensional Reconstruction

The 3D reconstruction of two histotripsy sites was successful. The first reconstruction (Figure 6A) showed an ovoid lesion resting on the porto-septal connective tissue and resting approximately 1 mm deep in the liver capsule without a connecting track (Figure 6A). The second reconstruction (Figure 6B) showed that the lesion was composed of several minute spots of hepatic plate injury aligned on an oblique axis to the liver capsule plane (Figure 6B) and scattered to a depth of 6 mm from the liver capsule.

## 4. Discussion

We have evaluated the effect of histotripsy in a viable perfused porcine liver ex vivo model with a histological evaluation of 15 excised lesions.

**Perilesional liver parenchyma**: The BH lesion was sharply demarcated and the hepatocytes and their supportive ECM facing the edge of the BH site appeared intact, histologically. We could not confirm, however, whether their enzymatic activity was preserved, using techniques such as the nicotinamide dinucleotide diaphorase (NADH-d) [16,25], because we used formalin-fixed and paraffin-embedded sections. Apoptosis has been described in the liver parenchyma surrounding areas of necrosis caused by in vivo radiofrequency ablation in pig livers [26] or by in vivo conventional thermal high intensity focused ultrasound (HIFU) in rabbit livers [27]. In both models, there was a time interval of a few days for the apoptotic process to develop fully, possibly due to reperfusion of the ablated tissue [26] and the effect of reactive oxygen species released locally [27].

**Portal tract involvement**: The border of the lesion extended to portal tracts, usually of small size, although, in one instance, a larger portal vein branch was involved and damaged, suggesting that vessels larger than the 300µm threshold, proposed by Khokhlova et al. [28], may be susceptible to histotripsy damage. Of note, rupture of vessels ranging in size from approximately 50 to 500 [29] and portal and hepatic vein thrombosis [30] have been observed in two in vivo histotripsy porcine models.

**Effect of BH on collagen and reticulin fibers**: Loss of SR-stained collagen birefringence has been observed in rodent aorta treated with laser welding [31] and in laser irradiated bovine tendons [32], but, to our knowledge, it is a novel finding in BH sites in pig liver. The loss of the reticulin framework inside the BH site indicates that BH does not affect the cellular component exclusively but it disrupts the ECM scaffolding of the hepatic plates. This is in line with the findings by Macoskey et al. [18], who showed a progressive deterioration of the ECM with increasing number of histotripsy pulses and followed by a loss of the cellular component. The presence of type D lesions away from the main histotripsy site are similar to the narrow regions of thermal damage described by Khokhlova et al. [6] in an in vivo porcine liver model. This observation suggests that thermal energy may be transmitted along connective tissue septa, a feature previously reported using liver microwave ablation [33]. However, it is possible that the margin of the cellular liquefactive area reaches these points in the deeper parts of the specimen that are not included in the planes of the sections examined. Regardless of the precise mechanism involved, the histotripsy track releases sufficient energy to break the connective tissue septa present physiologically in porcine liver, and the same effect would be expected on the pathological septa forming in chronically diseased hepatic tissue in humans as part of fibrosis progression.

**Liver capsule**: The BH site was subcapsular in some instances, but the liver capsule remained intact in our series despite the direct contact with the BH probe. The presence of an intact capsule is likely to reduce the impact of focal damage to vessels or biliary radicals following in vivo histotripsy, as any bleeding or bile leakage would be contained.

**Limitations**: There are technical limitations that need to be considered when interpreting our results. The specimens were obtained as part of a protocol that also involved cell harvesting from the histotripsy sites. The aspiration of the lesion liquid center or cell analysis may have partially affected the lesion appearances, particularly the formation of cavity-like spaces in some of the specimens. It is therefore not possible to confirm or exclude whether these cavities are due to boiling bubble nucleation or cavitation.

Our study design did not allow us to make a precise and reliable quantitative correlation between the type and extent of tissue damage and the BH parameters, including the effect of perfusion. The value of this histological review is to give information about the type of tissue damage observed, lesional boundaries, and the relationship with nearby hepatic parenchyma, portal structures, and connective tissue septa. The lesional area may not be fully represented in the two-dimensional assessment performed in each specimen.

**Three-dimensional reconstruction**. The purpose of the 3D reconstruction of two additional specimens was to address this limitation. Our two models showed that 3D reconstruction is feasible and gives valuable insight into the size, shape, and localization of the BH lesional site and its precise relationships to ambient anatomical structures. The first model, in particular (Figure 5a), shows that the lesional site rests against a portal tract and fibrous septa, suggesting that the connective tissue interface could reflect at least part of the energy. Further work is necessary to confirm this observation, but future studies aimed at refining the BH protocol to optimize the size, location, and nature of the BH lesions should consider the use of 3D reconstruction and correlate the 3D appearance, size, and location of the BH lesions with the BH procedure ultrasound parameters.

## 5. Conclusions

In summary, we describe in detail the effects of BH on a series of viable perfused porcine liver specimens. BH produces well-demarcated lesions, causing damage to both cellular and ECM components of the hepatic plates, and delivers sufficient energy to alter the configuration of collagen bundles, break ambient fibrous septa, and damage portal tract structures, including relatively large portal vein branches. The lesion is well demarcated with a very sharp border, where both the cellular and ECM components of the perilesional hepatic plates appear morphologically intact. The liver capsule is not affected by direct contact with the BH probe. A 3D reconstruction of BH sites is feasible and has great potential to correlate the BH parameters with the site, size, and nature of the BH liver lesions created.

## Figures and Tables

**Figure 1 bioengineering-10-00278-f001:**
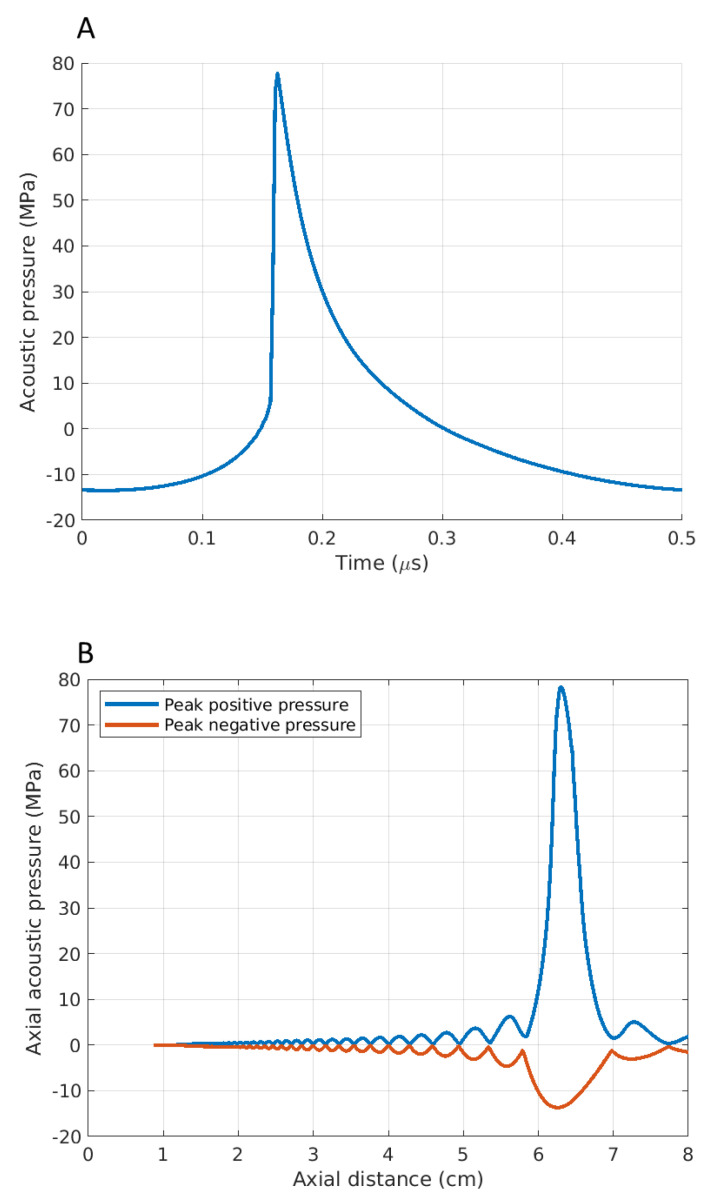
Simulated acoustic waveforms at (**A**) the HIFU focus on liver tissue and (**B**) axial pressure distribution.

**Figure 2 bioengineering-10-00278-f002:**
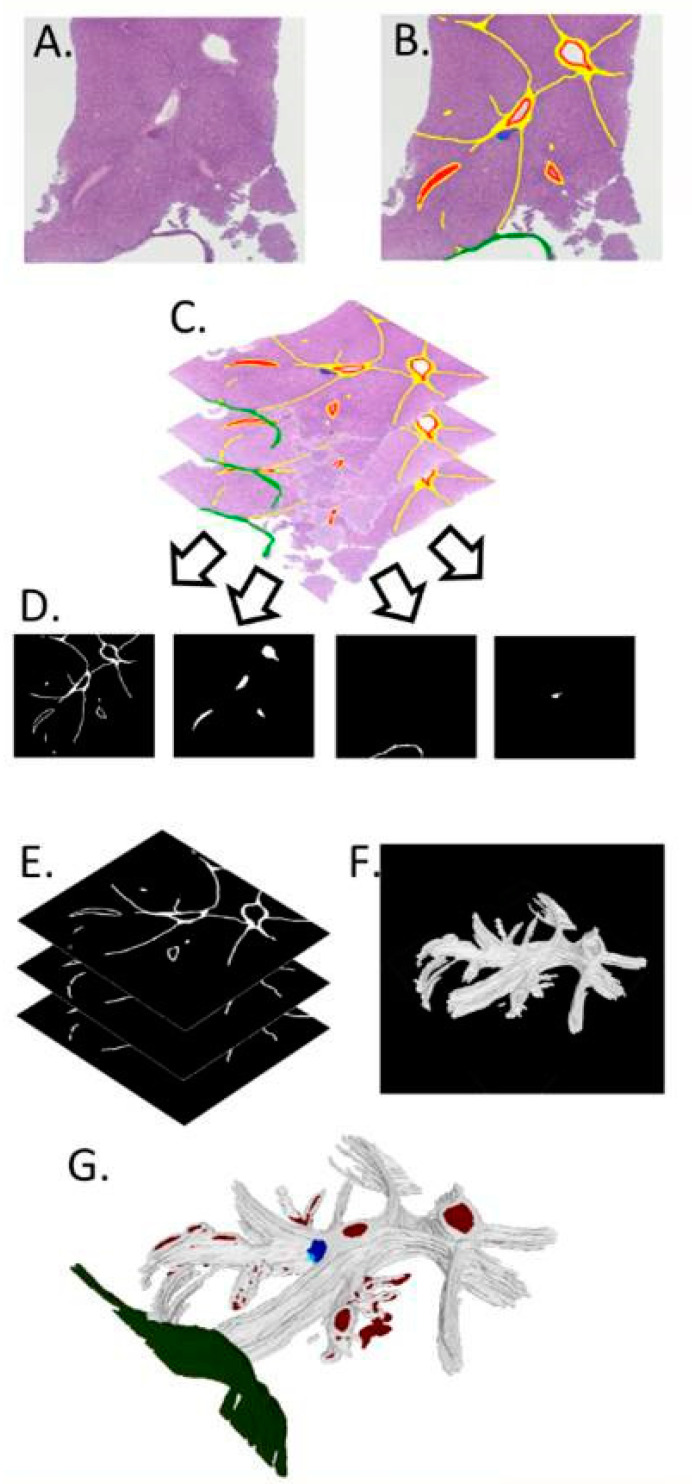
Three-dimensional reconstruction steps from serial sections and image acquisition (**A**), identification of structures of interest (**B**), registered images (**C**), visualization of two-dimensional binary matrices (**D**), three-dimensional matrix formation (**E**), creation of a mesh over the voxels (**F**), and final three-dimensional mesh visualization (**G**).

**Figure 3 bioengineering-10-00278-f003:**
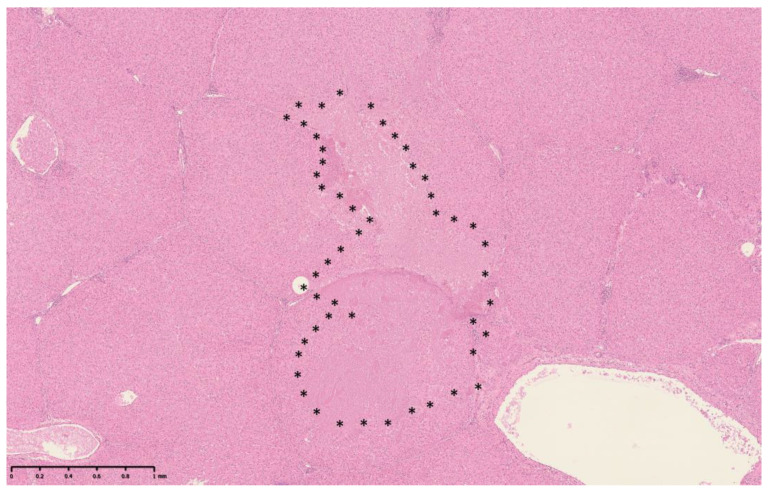
An example of a well-demarcated BH lesional site, measuring at 2.34 mm by 1.19 mm in this plane of section (scale bar shown), delineated by asterisks, and composed of type A and type B lesions, with minimal formation of cavities. The lesional site spreads over adjacent lobules and is not demarcated by interlobular septa (H&E). The cellular contents appear not to be fully aspirated at the time of procedure.

**Figure 4 bioengineering-10-00278-f004:**
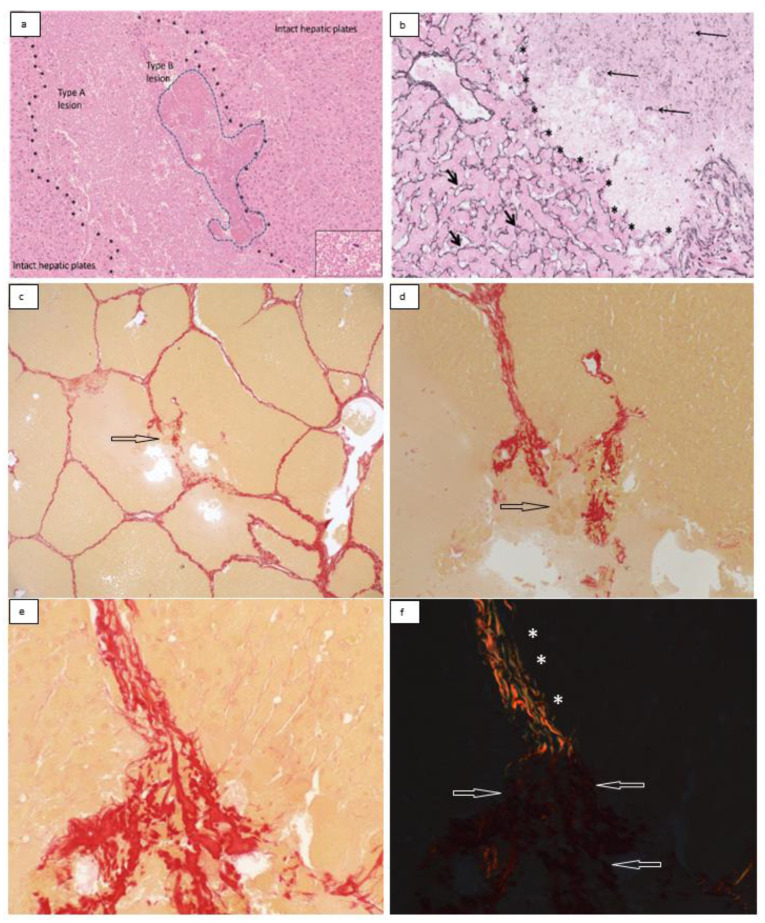
(**a**) H&E: type A and B lesions. Nuclear debris and scattered nuclei were present in type A lesions (**a,** inset). (**b**) Reticulin stain: loss and fragmentation of reticulin fibers (long thin arrows) at lesional site. Please note sharp transition (asterisks), with nearby preserved hepatic plates showing an intact reticulin framework (short arrows). (**c**–**f**) Picro–Sirius red stain. (**c**,**d**) A broken interlobular septum is shown at lower (**c**, arrow) and higher (**d**) magnification. (**e**,**f**) show retention (asterisks) and loss (arrows) of collagen birefringence in type C and type D lesions, respectively.

**Figure 5 bioengineering-10-00278-f005:**
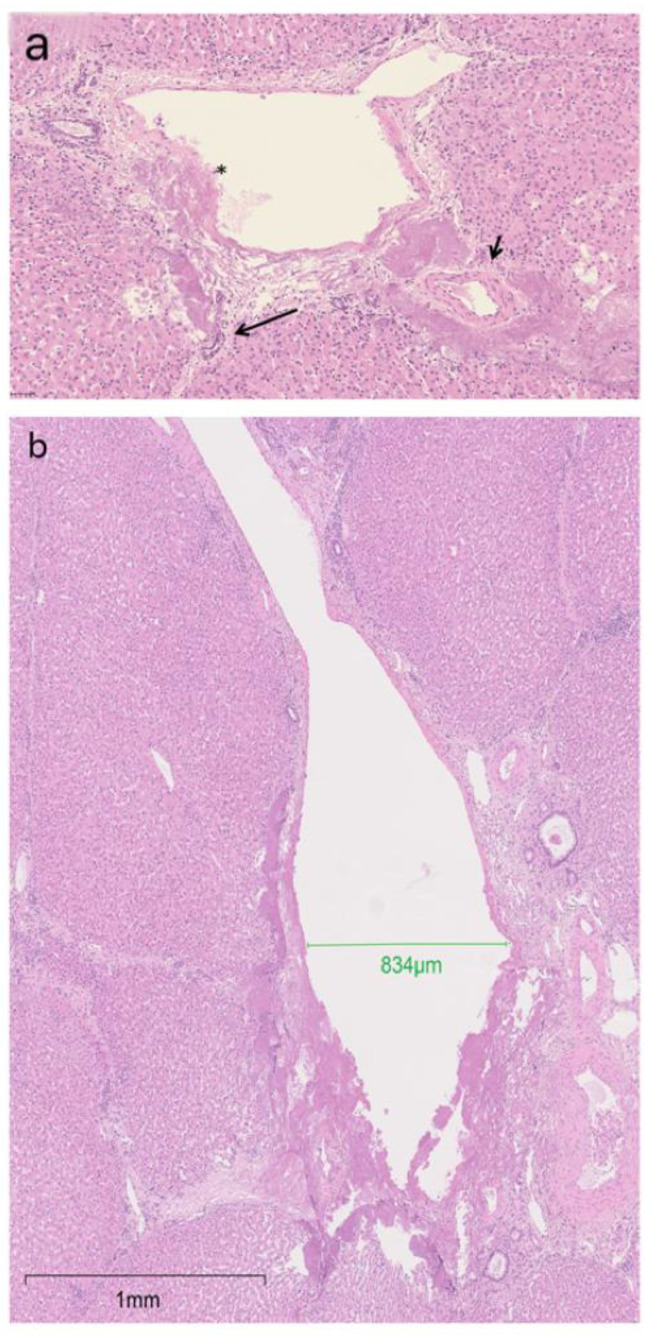
(**a**) BH lesion extending to a nearby portal tract with injury to portal vein (asterisk), bile duct (long arrow), and encasement of an arterial branch (short arrow) (H&E). (**b**) Injury to a portal vein that is 834 µm in diameter (green bar).

**Figure 6 bioengineering-10-00278-f006:**
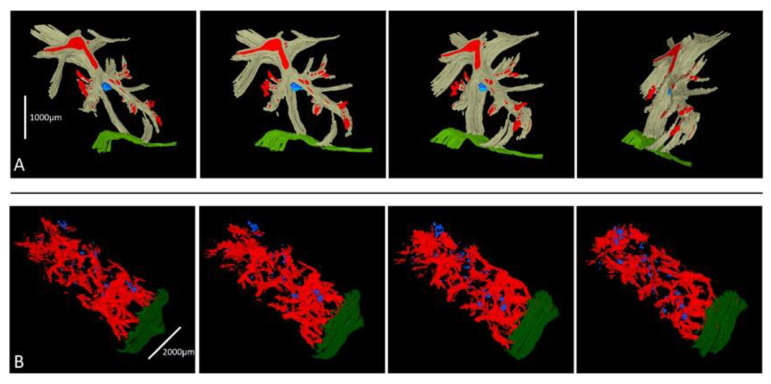
Three-dimensional reconstruction of two HT lesions. (**A**) The first HT lesion is shown as a blue nodular area resting against the convergence of two fibro-connective vascularized (red) septa (grey) and well away from the capsule (green). The second HT lesion (**B**) is shown as multiple blue lesional foci aligned on an oblique axis to the liver capsule plane (green). Vasculature is in red.

## Data Availability

Not applicable.

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
