# Peer review of "Ultrasound Histotripsy on a Viable Perfused Whole Porcine Liver: Histological Aspects, Including 3D Reconstruction of the Histotripsy Site"

_bioengineering, 2023, doi:10.3390/bioengineering10030278_

Round 1

Reviewer 1 Report

In this paper, the influence of ultrasonic tissue comminution technology and the feasibility of 3D reconstruction of tissue comminution site were verified by using live perfusion of in vitro pig livers as samples.

The layout of the paper is reasonable, the language organization is clear, and the description of the experimental method BH (boiling tissue crushing), the 3D reconstruction steps and the treatment of the sample is very detailed, which can be easily understood by the readers.

It should be noted that the main content of the experimental design only includes the control of tissue lesion samples and the 3D reconstruction effect of the crushing site, and the description of the possible influence of ultrasonic tissue crushing on the subject of the study needs to be supplemented. In the process of the experimental design, these possible influences can also be considered to include, which will enrich the content of the article. The material used in the experiment was 5 fresh pig livers, and the control group was small. More repetitions could be made to make the results more reliable. In the article layout, you can display the chart in the results section.

Author Response

In this paper, the influence of ultrasonic tissue comminution technology and the feasibility of 3D reconstruction of tissue comminution site were verified by using live perfusion of in vitro pig livers as samples.

The layout of the paper is reasonable, the language organization is clear, and the description of the experimental method BH (boiling tissue crushing), the 3D reconstruction steps and the treatment of the sample is very detailed, which can be easily understood by the readers.

It should be noted that the main content of the experimental design only includes the control of tissue lesion samples and the 3D reconstruction effect of the crushing site, and the description of the possible influence of ultrasonic tissue crushing on the subject of the study needs to be supplemented. In the process of the experimental design, these possible influences can also be considered to include, which will enrich the content of the article. The material used in the experiment was 5 fresh pig livers, and the control group was small. More repetitions could be made to make the results more reliable. In the article layout, you can display the chart in the results section.

Response 1: Thank you for the comments. Although we have only used 5 pig livers in this experiment, the lesion numbers generated and the degree of inter-lesional variability reassure us that the experiment numbers are adequate and representative. Whilst we acknowledge the limitations of the current study, the aim of this piece of work was to demonstrate the character of the lesions histologically as part of a broader and bigger scope of work and assess the lesion site for its suitability in cell transplantation. Hence the 3D reconstruction to assess the nidus and visualise the host environment for implantation of the cells.

Reviewer 2 Report

There are some scientific interests in this manuscript. But the overall design and writing levels of the experiments should be improved significantly. For example, in the title: “Ultrasound histotripsy on a viable perfused whole porcine liver: histological aspects including 3D reconstruction of the histotripsy site, it is easy to be considered as “Ultrasound histotripsy of a porcine liver and 3D constructing the lesion with new tissues. There are many writing errors throughout the manuscript, especially the units. The abbreviations are extremely confusion. Most of the original names are lacking. This reviewer suggests to revise it carefully. 

Author Response

There are some scientific interests in this manuscript. But the overall design and writing levels of the experiments should be improved significantly. For example, in the title: “Ultrasound histotripsy on a viable perfused whole porcine liver: histological aspects including 3D reconstruction of the histotripsy site”, it is easy to be considered as “Ultrasound histotripsy of a porcine liver and 3D constructing the lesion with new tissues”. There are many writing errors throughout the manuscript, especially the units. The abbreviations are extremely confusion. Most of the original names are lacking. This reviewer suggests to revise it carefully. 

Response 1: Thank you for the comments. We have revised the manuscript accordingly with suggested comments. We have made changes to the units where necessary to make them uniform and more accessible for the reader to follow. Additionally, the manuscript has been revised for any spelling or grammatical errors. Also, we have re-checked our numerical values to ensure its correct. Figure one has been reproduced to reflect the simulated acoustic waveform and axial pressure distribution accurately.